# REVISITING EMBEDDINGS FOR GRAPH NEURAL NETWORKS

## ABSTRACT

Current graph representation learning techniques use Graph Neural Networks (GNNs) to extract features from dataset embeddings. In this work, we examine the quality of these embeddings and assess how changing them can affect the accuracy of GNNs. We explore different embedding extraction techniques for both images and texts; and find that *the choice of embedding biases the performance of different GNN architectures and thus the choice of embedding influences the selection of GNNs regardless of the underlying dataset*. In addition, we only see an improvement in accuracy from some GNN models compared to the accuracy of models trained from scratch or fine-tuned on the underlying data without utilising the graph connections. As an alternative, we propose ***Graph-connected Network (GraNet)*** layers to better leverage existing unconnected models within a GNN. Existing language and vision models are thus improved by allowing neighbourhood aggregation. This gives a chance for the model to use pre-trained weights, if possible, and we demonstrate that this approach improves the accuracy compared to traditional GNNs: on Flickr_v2, GraNet beats GAT2 and GraphSAGE by 7.7% and 1.7% respectively.

## 1 INTRODUCTION

Graph Neural Networks (GNNs) have been successful on a wide array of applications ranging from computational biology (Zitnik & Leskovec, 2017) to social networks (Hamilton et al., 2017). The input for GNNs, although sourced from many different domains, is often data that has been preprocessed to a computationally digestible format. These digestible formats are commonly known as embeddings.

Currently, improvements made to GNN architecture are tested against these embeddings and the state of the art is determined based on those results. However, this does not necessarily correlate with the GNNs accuracy on the underlying dataset and ignores the influence that the source and style of these embeddings have on the performance of particular GNN architectures. To test existing GNN architectures, and demonstrate the importance of the embeddings used in training them, we provide three new datasets each with a set of embeddings generated using different methods.

We further analyse the benefit of using GNNs on fixed embeddings. We compare GNNs to standard models that have been trained or fine-tuned on the target raw data; these models treat each data point as unconnected, ignoring the underlying graph information in data. This simple unconnected baseline surprisingly outperforms some strong GNN models. This then prompts the question: *Will mixing the two approaches unlock the classification power of existing unconnected models by allowing them to utilize the graph structure in our data?*

Based on the question above, we propose a new method of mixing GNNs with unconnected models, allowing them to train simultaneously. To achieve this we introduce a variation of the standard message passing framework. With this new framework a subset of the unconnected model's layers can each be graph-connected – exploiting useful graph structure information during the forward pass. We demonstrate that this new approach improves the accuracy of using only a pre-trained or fine-tuned model and outperforms a stand-alone GNN on a fixed embedding.

We call this new approach **GraNet** (Graph-connected Network), and in summary, this paper has the following contributions:

- We provide new datasets and a rich set of accompanying embeddings to better test the performance of GNNs.
- We empirically demonstrate that only some existing GNNs improve on unconnected model accuracy and those that do vary depending on the embeddings used. We urge that unconnected models be used as a baseline for assessing GNN performance.
- We provide a new method, named GraNet, that combines GNNs and models (fine-tuned or trained from scratch) to efficiently exploit the graph structure in raw data.
- We empirically show that GraNet outperforms both unconnected models (the strong baseline) and GNNs on a range of datasets and accompanying embeddings.

## 2 RELATED WORK

**Graph Augmented Networks** Chen et al. (2021) introduce Graph-augmented MultiLayer Perceptrons (GA-MLPs) as a simplified alternative to Graph Neural Netwroks (GNNs). These models involve a two step process - augmenting the node features of the graph based on the topology and then using these node features applying a learnable function at the node level. This allows a fixed graph operator and two sets of MultiLayer Perceptrons (MLPs) be used to extract features from the graph. This approach is related to similar simplified GNN techniques (Wu et al., 2019; Nt & Maehara, 2019). The paper proves that this simplified approach is not as expressive as standard GNNs when looking at the Weisfeiler-Lehman test for distinguishing non-isomorphic graphs. This suggests that GNNs are well suited for inferring information based on graph structure but the paper does not comment on which approach is best in practice. We differ in our approach to augmenting networks with graph structure by using existing GNNs and do not attempt to simplify the network. We do provide a graph-connected MLP but this looks at adding message passing to MLPs rather than separte funcitons on the graph data.

**Effect of training on GNN performance** Shchur et al. (2018) look at the effect of hyperparameters and training in GNNs to show that these have dramatic effect on model ordering. Simply changing the split on a dataset caused large changes in accuracy and which GNN performed best, even though the hyperparameters of the GNNs remained constant. We show similar large difference when considering different embeddings with the same splits across embeddings.

**Ablation studies on GNNs** Further to these discoveries Nt & Maehara (2019) demonstrate that GNNs only utilise the graph structure to de-noise already highly informative features. They go as far as to demonstrate in certain conditions GNNs and MLPs perform the same. Chen et al. (2019) demonstrate that linearising the graph filter stage of GNNs does not hinder but actually increases the performance. Similarly Wu et al. (2019) simplify GNNs by removing non-linearity between layers allow for pre-computing the $k$ message passes. This reduces graph representation learning to a simple linear regression. In all of these cases they demonstrate that the major contribution of GNNs is in their graph structure capabilites. We do not analyse these aspects but look at how this capability can be used in existing unconnected networks.

## 3 BACKGROUND

Table 1: An overview of popular datasets

| Name | Info Source | Classes | Feature Length | Embedding |
|---|---|---|---|---|
| Amazon (Shchur et al., 2018) | Text | 10, 8 | 767, 745 | Bag of Words |
| AmazonProducts (Zeng et al., 2019) | Text | 107 | 200 | 4-gram with SVD |
| Flickr (Zeng et al., 2019) | Image | 7 | 500 | Bag of Words |
| Reddit (Zeng et al., 2019; Hamilton et al., 2017) | Text | 41 | 602 | Avg. GloVe vectors |
| Cora (Kipf & Welling, 2017) | Text | 7 | 1,433 | Bag of Words |
| CiteSeer (Kipf & Welling, 2017) | Text | 6 | 3,703 | Bag of Words |
| PubMed (Kipf & Welling, 2017) | Text | 3 | 500 | Bag of Words |

We compare our new method (GraNet) against some standard Graph Neural Networks to demonstrate the improvements that GraNet makes in classifying datasets.

Kipf & Welling (2017) introduce GCN (Graph Convolutional Networks) – a method of applying convolutional layers from CNNs to graph neural networks. It focuses on spectral filters applied to the whole graph structure rather than at the node level.

Hamilton et al. (2017) introduce the GraphSAGE model which builds on prior work from GCN focusing on individual node representations. This gives rise to the iterative message passing process on the node level. Though simpler than newer models we find that this approach, when given the right embedding style, can outperform some recently published GNNs.

Veličković et al. (2018) introduce the idea of graph attention which alters how a node aggregates its neighbours representation. This adds an additional attention mechanism to discern which aspects of the node representations in a nodes neighbourhood are important at a given layer.

Brody et al. (2021) provide a more attentive version based on the graph attention system introduced in Veličković et al. (2018). We base the graph attention mechanism used in our GraNet models on this improved version of graph attention. We provide both versions of graph attention in our results to compare to our new approach.

## 3.1 Notations

**Graph Data** Let $\mathcal{G}(\mathbb{V}, \mathcal{E}, \boldsymbol{X})$ denote a graph where $\mathbb{V} = \{\boldsymbol{v}_1, \boldsymbol{v}_2, ..., \boldsymbol{v}_n\}$ is the set of nodes and $N = |\mathbb{V}|$ is the number of nodes in the graph, $\mathcal{E}$ is the set of edges between nodes in $\mathbb{V}$ such that $\boldsymbol{e}_{i,j} \in \mathcal{E}$ denotes a directed connection from node $\boldsymbol{v}_i$ to node $\boldsymbol{v}_j$, $\boldsymbol{e}_{i,j}$ may itself be a feature vector. We say each node $\boldsymbol{v}_i$ has a neighbourhood $\mathbb{N}_i$ such that $\boldsymbol{v}_j \in \mathbb{N}_i \iff \boldsymbol{e}_{j,i} \in \mathcal{E}$ and we say that $\boldsymbol{v}_j$ is a neighbour node to $\boldsymbol{v}_i$. Where $\boldsymbol{X}$ is the raw data matrix where $\boldsymbol{X}_{:,i} = \boldsymbol{x}_i$ where $\boldsymbol{x}_i$ is the feature vector for node $\boldsymbol{v}_i$.

**Embeddings** There normally exists a transformation function, $f_e$, to project the raw data to a more compact feature $\boldsymbol{X}_e$ space such that $\boldsymbol{X}_e = f_e(\boldsymbol{X})$

For instance, we can transform a set of images ($\boldsymbol{X} \in \mathbb{R}^{N \times C \times H \times W}$, where $C$, $H$ and $W$ are the number of channels, width and height of an image) to 1D features ($\boldsymbol{X}_e \in \mathbb{R}^{N \times F}$, where $F$ denotes the feature dimension). In this case, we have an embedding function $f_e : \mathbb{R}^{N \times C \times H \times W} \to \mathbb{R}^{N \times F}$ for the dimensional reduction.

This paper puts a special focus on the design of $f_e$, and reveals later how the design choice of $f_e$ can influence the performance of GNN models without making any changes to the underlying data $\mathcal{G}(\mathbb{V}, \mathcal{E}, \boldsymbol{X})$. An overview of popular datasets, and the embeddings that they use, is presented in Table 1. We see that the popular graph datasets (Zeng et al., 2019; Kipf & Welling, 2017; Shchur et al., 2018; Hamilton et al., 2017) focus heavily on Bag of Words (BoW) and word vectors. This implies that current GNNs are being tested on and designed for a very narrow class of embedding styles. A more detailed discussion is available in Appendix A.

**Graph Neural Networks** Current GNNs can be thought of as *message passing* layers, the $l$-th layer can be represented as

$$\boldsymbol{h}_i^l = \gamma_{\boldsymbol{\theta}_\gamma} \left( \boldsymbol{h}_i^{l-1}, \psi_{j \in \mathbb{N}(i)} \left( \phi_{\boldsymbol{\theta}_\phi} \left( \boldsymbol{h}_i^{l-1}, \boldsymbol{h}_j^{l-1}, \boldsymbol{e}_{j,i} \right) \right) \right) \tag{1}$$

where $\psi$ is a differentiable aggregation function and $\gamma_{\boldsymbol{\theta}_\gamma}$ and $\phi_{\boldsymbol{\theta}_\phi}$ represent differentiable functions with trainable parameters $\boldsymbol{\theta}_\gamma$ and $\boldsymbol{\theta}_\phi$ respectively. $\boldsymbol{h}_i^l$ is the node representation of $v_i$ at layer $l$, with $\boldsymbol{h}_i^0 = f_e(\boldsymbol{x}_i)$.

We focus on the improved graph attention mechanism (Brody et al., 2021) when looking at potential GNNs for GraNet. Using this new notation we can formulate it as such

$$\alpha_{ij} = \frac{exp(\boldsymbol{a}^T \text{LeakyReLU}(\boldsymbol{\theta}[\boldsymbol{h}_i || \boldsymbol{h}_j]))}{\sum_{k \in \mathbb{N}_i} exp(\boldsymbol{a}^T \text{LeakyReLU}(\boldsymbol{\theta}[\boldsymbol{h}_i || \boldsymbol{h}_j]))} \tag{2}$$

where $\boldsymbol{a}$ is a learnable parameter representing the attention of the network. Further discussion of graph attention and why this is useful in GraNet is available in Appendix C.

**Pre-trained models** are specific neural network architecturwa that have been trained on a dataset $\mathcal{D}$ for a specific task, this could be ImageNet classification (Deng et al., 2009) for vision networks (He et al., 2016) or the entire English Wikipedia for language models (Liu et al., 2019). These networks therefore have *pre-trained* weights $\boldsymbol{\theta}$ that can be loaded into the model for further training or evaluation.

We denote these pre-trained models as $f_{\boldsymbol{\theta}}$ that is parameterised by weights $\boldsymbol{\theta}$. We say that a pre-trained model has a set of functions $\{f^1_{\boldsymbol{\theta}_1}, f^2_{\boldsymbol{\theta}_2}, ..., f^M_{\boldsymbol{\theta}_M}\}$ for an $M$-layer model, where $f^i_{\boldsymbol{\theta}_i} : \mathbb{R}^F \to \mathbb{R}^{F'}$ and $f^{i+1}_{\boldsymbol{\theta}_{i+1}} : \mathbb{R}^{F'} \to \mathbb{R}^{F''}$ and $F$ is the feature dimension. A pass through a single layer, $l$, of a network would be $f^l_{\boldsymbol{\theta}_l}(\boldsymbol{x})$, shorthand for $f^l(\boldsymbol{x}; \boldsymbol{\theta}_l)$. If we concatenate these layers to form a full pass through the network, we obviously have $f_{\boldsymbol{\theta}}(\boldsymbol{x}) = f^M_{\boldsymbol{\theta}_M}(...f^2_{\boldsymbol{\theta}_2}(f^1_{\boldsymbol{\theta}_1}(\boldsymbol{x})))$

**Fine-tuning** is therefore adapting $\boldsymbol{\theta}$ to a new dataset $\mathcal{D}'$ which is related to $\mathcal{G}(\mathbb{V}, \mathcal{E}, \boldsymbol{X})$ as illustrated in Section 3.1. In a more standard setup, the target dataset we fine-tune to has the same underlying data-structure as the pre-training dataset, for instance, they might both be images, but the target dataset is a different type of classification. This may involve adding, removing or altering specific layers within the model or simply retraining the model with different labels on $\mathcal{D}$.

If the architecture of a pre-trained model is altered then a new weight matrix $\boldsymbol{\theta}'$ must be created from $\boldsymbol{\theta}$ by adding, removing or reshaping weights.

**Freezing** layers is the process whereby a selection of weights $\boldsymbol{\theta}_f \subset \boldsymbol{\theta}$ do not have gradients and thus do not change during back-propagation.

These ideas allow us to alter these pre-trained models to use information about the graph connections whilst utilising their pre-trained weight $\boldsymbol{\theta}$.

**Blending and Fine-tuning** models is therefore the process of using existing models, $f_{\boldsymbol{\theta}_f}$ and $g_{\boldsymbol{\theta}_g}$, with defined set of layers, $\{f^1_{\boldsymbol{\theta}_{f1}}, f^2_{\boldsymbol{\theta}_{f2}}, ..., f^N_{\boldsymbol{\theta}_{fN}}\}$ and $\{g^1_{\boldsymbol{\theta}_{g1}}, g^2_{\boldsymbol{\theta}_{g2}}, ..., g^M_{\boldsymbol{\theta}_{gM}}\}$, and creating a new model, $h_{\boldsymbol{\theta}_{f,g}}$, such that $h^i_{\boldsymbol{\theta}_{f,g}} = f^j_{\boldsymbol{\theta}_f} \circ g^k_{\boldsymbol{\theta}_g}$. We can use pre-trained weights $\boldsymbol{\theta}_f$ or $\boldsymbol{\theta}_g$ and/or freeze either model, and where one of these models is a GNN we say this is fine-tuning on a graph dataset.

**Unconnected models** are models that, unlike GNNs, do not use any information about graph connections within a dataset. These are trained on datasets that are not graph-connected where each datapoint is consider isolated. We focus on complex unconnected models with multiple layers, such as vision networks, which we call *large models*. Due to training cost we use *pre-trained large models*.

## 4  METHOD

Our proposed method of converting standard neural network models into graph-connected models blends the two networks. This approach can easily be broken down into individual layers. Taking $f^l_{\boldsymbol{\theta}_l}$ as the $l$-th layer in a standard model, $f_{\boldsymbol{\theta}}$, where we may be given pre-trained weights $\boldsymbol{\theta}$ we can describe this new layer by reformulating Equation (1) as such

$$\boldsymbol{h}^l_i = \gamma_{\boldsymbol{\theta}_\gamma}\left(\boldsymbol{h}^{l-1}_i, \psi_{j \in \mathbb{N}(i)}\left(\phi_{\boldsymbol{\theta}_\phi}\left(f^{l-1}_{\boldsymbol{\theta}_{l-1}}\left(\boldsymbol{h}^{l-1}_i\right), f^{l-1}_{\boldsymbol{\theta}_{l-1}}\left(\boldsymbol{h}^{l-1}_j\right), \boldsymbol{e}_{j,i}\right)\right)\right) \tag{3}$$

where $\mathbf{h}^0_i = \mathbf{x}_i$ rather than applying an embedding function.

Figure 1 is a representation of Equation (3) specifically in the case where the pre-trained model is a CNN and $(f^{l-1}_{\boldsymbol{\theta}_{l-1}})$ is a convolutional layer. The light blue regions perform the standard convolution feature extraction, these extracted feature maps are then adjusted, in the light orange region, by a Message Passing layer with graph attentions. These two representations are then combined.

The light blue regions and resulting channel stacks, $A_i \in \mathbb{R}^{1 \times C_l \times H_l \times W_l}$ through $A_n \in \mathbb{R}^{1 \times C_l \times H_l \times W_l}$, represent the forward pass through a single CNN layer $f^{l-1}_{\boldsymbol{\theta}_{l-1}}$. $A_i$ represents the forward pass of the current node $\boldsymbol{h}_i$ and $A_{\{j,...,n\}}$ represent the forward pass of the neighbours of $\boldsymbol{h}_i$. The light orange region and resulting channel stacks, $\mathcal{B}$, represent the graph-based Message

Figure 1: Graphical representation of how our proposed GraNet layer operates for image networks

Passing stage where the new representations are altered ($\phi_{\boldsymbol{\theta}_\phi}$), aggregated ($\psi$) and finally combined with the current node representation ($\gamma_{\boldsymbol{\theta}_\gamma}$), following the description in Equation (3).

## 4.1 GRANET FOR FLICKR_V2

Interconnecting every layer in a large pre-trained model is computationally intensive. Therefore, rather than interconnect every single layer in a pre-trained network $f_{\boldsymbol{\theta}}$ we can graph-connect only the final layer. We thus split the model into two portions: the first set of unconnected layers and the final GraNet layer.

We can therefore look at equation Equation (3) and see in this case that we partially carry out the forward pass of $f_{\boldsymbol{\theta}}$ which we will denote as $f_e$ and then carry out a forward pass through a GraNet layer. This allows us to ignore all the steps involved in applying $f_e$ and focus on the final GraNet layer. Denoting this final GraNet as $g$, and output classification $y$, we achieve the following equation

$$y = g(f_e(\boldsymbol{X})) \tag{4}$$

$f_e(\boldsymbol{X})$ is the same as described in Section 2 and indeed if we were to freeze $f_{\boldsymbol{\theta}}$ this would be equivalent to training GraNet with a single layer on the embeddings created by embedding function $f_e$. So instead we also allow $f_e$ to train thus fine-tuning the weights $\boldsymbol{\theta}$ of $f_{\boldsymbol{\theta}}$. This indirectly allows $f_{\boldsymbol{\theta}}$ to learn the graph structure by providing $g$ with better embeddings.

As GraphSAGE performed the best on ResNet embeddings we use this as our GNN, $g$. As $f_{\boldsymbol{\theta}_f}$ is already pre-trained but $g_{\boldsymbol{\theta}_g}$ is not fine-tuning the model produces poor results. We therefore initially train $g_{\boldsymbol{\theta}_g}$ on a frozen $f_{e\boldsymbol{\theta}_f}$, after this short training period we unfreeze $f_{e\boldsymbol{\theta}_f}$ allowing both to train fine-tuning the weights $\boldsymbol{\theta}_g$ and $\boldsymbol{\theta}_f$.

## 4.2 GRANET FOR AMAZON

In the case of the Amazon dataset we found that Bag of Words embedding performed the best. As this does not have an associated pre-trained model we design a multi-layer perceptron (MLP) to compare against. We then convert all the layers within the MLP to GraNet layers. *This method is therefore not fine-tuned but trained from initialised weights.*

We also find that GAT (Brody et al., 2021; Veličković et al., 2018) models perform the best on this task. We therefore use graph attention message passing as shown in Equation (2) for our GraNet model. Keeping in line with Equation (3) the new graph attention mechanism becomes

$$\alpha_{ij} = \frac{exp(\boldsymbol{a}^T\text{LeakyReLU}([f^l_{\boldsymbol{\theta}_l}(\boldsymbol{h}_i)||f^l_{\boldsymbol{\theta}_l}(\boldsymbol{h}_j)]))}{\sum_{k\in\mathbb{N}_i} exp(\boldsymbol{a}^T\text{LeakyReLU}([f^l_{\boldsymbol{\theta}_l}(\boldsymbol{h}_i)||f^l_{\boldsymbol{\theta}_l}(\boldsymbol{h}_j)]))} \tag{5}$$

However, a single layer of an MLP is $f^l_{\boldsymbol{\theta}_l}(\boldsymbol{x}) = \boldsymbol{\theta}_l\boldsymbol{x}$. This would therefore mean that Equation (6) becomes

$$\alpha_{ij} = \frac{exp(\boldsymbol{a}^T \text{LeakyReLU}([\boldsymbol{\theta}_l \boldsymbol{h}_i || \boldsymbol{\theta}_l \boldsymbol{h}_j]))}{\sum_{k \in \mathbb{N}_i} exp(\boldsymbol{a}^T \text{LeakyReLU}([\boldsymbol{\theta}_l \boldsymbol{h}_i || \boldsymbol{\theta}_l \boldsymbol{h}_j]))} \tag{6}$$

This is very similar to Equation (2) with the only difference being when we apply the weight matrix, concatenation and attention mechanism. We find this model behaves the same as GAT2 and therefore, given the small size tried a different approach as a comparison.

Rather than use the vector parameter $\boldsymbol{a}$ we introduce a linear function $a_{\boldsymbol{\theta}_a} : \mathbb{R}^{2F} \to \mathbb{R}^{F'}$ that takes the LeakyReLU of the concatenation of a node, $f_{\boldsymbol{\theta}_i}(\boldsymbol{h}_i)$, and its neighbour, $f_{\boldsymbol{\theta}_i}(\boldsymbol{h}_i)$, as input. The result is a feature vector in a new feature space, though in our case we have $F' = F$. This allows more complex interactions between the node representations to be exploited by our attention. This approach is too costly to apply to a pre-trained CNN as the size of $\boldsymbol{\theta}_a$ is far too large.

## 5 EXPERIMENTAL SETUP

For all test results we run 3 train-test runs each with a different random seed and take the arithmetic mean and include the standard deviation. The three random seeds are the same for every entry in the table for fair comparison. The architecture used for each benchmark GNN is identical across all datasets and embeddings. Training takes 300 epochs of training unless otherwise stated.

Where a Graph Neural Network (GNN) is used as part of a GraNet model (as described in Section 4) the same sampler is used for consistency. The specific architectures used are designed using the hyperparameters in Zeng et al. (2019). In the case of GraphSAGE the learning rate was decreased to improve convergence. For specific details of each architecture and learning rate see Appendix E.

### 5.1 DATASETS

An overview of the datasets is provided below with the specific metrics for each dataset shown in Table 6. A more detailed discussion is available in Appendix B. *It is important to note that though these datasets mirror prior datasets due to the need for raw data we diverge from these datasets.* Therefore we do not make any direct comparisons to previous datasets though the results we achieve on our new datasets are on par with results seen in prior papers.

#### 5.1.1 FLICKR_V2

The underlying data $\boldsymbol{X}$ is raw images and so Convolutional Neural Networks (CNNs) are used as embedding functions. As a sample of existing pre-trained CNNs we use ResNet18, ResNet50 (He et al., 2016) and VGG16 (Simonyan & Zisserman, 2015). In all three cases we use the pre-trained models provided by TorchVision, using the feature vectors after the final pooling stage before the classification stage.

It is important to note that *there is no Bag of Words (BoW) embedding for Flickr_v2 because there is no sensible object that can be considered a "word" for raw image data.* The available Flickr (Zeng et al., 2019) uses BoW because the underlying data is image text descriptions not raw images. This limits how a GNN, or general neural network, can classify images as the images must first be processed to provide text descriptions.

#### 5.1.2 AMAZONELECTRONICS AND AMAZONINSTRUMENTS

The underlying data $\boldsymbol{X}$ is text so text classification transformers such as RoBERTa (Liu et al., 2019) are ideal, specifically we use pre-trained RoBERTa. We extract three different embeddings from RoBERTa using the pre-trained model provided by fairseq toolkit (Ott et al., 2019). The first is the byte pair tokenisation used by RoBERTa, the second is the feature extraction provided by fairseq which occurs after RoBERTa's transformer heads and before classification, and the final is the feature vector present before the last fully connected layer. Due to restrictions in the token size for RoBERTa we remove all nodes that have reviews with greater than 512 tokens.

We also provide the standard Bag of Words embedding as in the case of text classification this is a common embedding practice, keeping in line with prior datasets (Kipf & Welling, 2017; Zeng et al., 2019) we use the top 500 words to create our Bag of Word embeddings.

# 6 EVALUATION

## 6.1 RE-EVALUATING EMBEDDINGS

**Language Tasks** Table 2 demonstrates that the particular embedding function used determines which GNN model performs best on the dataset. For instance, GAT2 has the best performance if the data is embedded as BoW (Bag of Words), but performs poorly on other embeddings generated from RoBERTa. GraphSAGE, in contrast, performs poorly on BoW but shows a good performance otherwise. Thus which embeddings are used when comparing models has a large effect on which model appears to be the better model. We see the same effect in Table 3. Comparing the two tables we see that in the case of the RoBERTa Encoded and RoBERTa embeddings the order of the models changes, this likely comes from the training effects described in Shchur et al. (2018).

Table 2 also signifies the importance of good embeddings as in this case BoW is better than RoBERTa. The complexity of the embeddings does not necessarily improve the efficiency of the classification. This ties into the results of Nt & Maehara (2019) as the BoW is more informative of the classification containing the label words in the BoW vector. What is also important to note is that when looking at the performance of the GNN against the unconnected model ($\Delta \uparrow$) we see consistency in the difference across the RoBERTa encodings. *This suggests that the "quality", denoted by how well the simple MLP performs, is a strong indicator of how well a model will perform, rather than the model architecture itself.*

We see an overall decrease in accuracy in Table 3 across the models but this attributed to the fact that there are more classes for the dataset. We also see that for RoBERTa embeddings the MLP performs poorly though it does occasionally improve on the simpler models, primarily GCN. With more resources it would be better suited to fine-tune RoBERTa (or some other language transformer) to our dataset and use this model as our unconnected model.

Table 2: Test accuracy on AmazonElectronics with different embeddings compared against a standard unconnected MLP model. The embedding styles are explained in Appendix B, Table 6.

| Model | Embedding styles | | | |
| --- | --- | --- | --- | --- |
| | Bag of Words | Byte Pair | RoBERTa Encoded | RoBERTa |
| Unconnected MLP | 71.6% (+0.0) | 21.6% (+0.0) | 55.8% (+0.0) | 51.9% (+0.0) |
| GCN | 69.1% (-2.5) | 21.7% (+0.1) | 22.7% (-33.1) | 22.3% (-29.6) |
| GAT | *81.1% (+10.5)* | 22.2% (+0.6) | 46.1% (-9.7) | 40.3% (-11.6) |
| GAT2 | **81.8% (+10.2)** | 22.2% (+0.6) | 41.8% (-14.0) | 35.7% (-16.2) |
| GraphSAGE (Random) | 71.3% (-0.3) | *26.8% (+5.2)* | *57.0% (+1.2)* | *53.7% (+1.8)* |
| GraphSAGE (Neighbour) | 76.4% (+4.8) | **40.4% (+20.8)** | **67.8% (+12.0)** | **66.4% (+12.5)** |

Table 3: Test accuracy on AmazonInstruments with different embeddings compared against a standard unconnected MLP model. Included is the difference $\Delta$ of each model to the unconnected MLP and the standard deviation of each result. The embedding styles are explained in Appendix B.

| Model | Embedding Styles | | | |
| --- | --- | --- | --- | --- |
| | Bag of Words | Byte Pair | RoBERTa Encoded | RoBERTa |
| Unconnected MLP | 66.1% (+0.0) | 21.0% (+0.0) | 43.9% (+0.0) | 39.8% (+0.0) |
| GCN | 64.0% (-2.1) | 20.8% (-0.2) | 20.4% (-23.5) | 20.4% (-19.4) |
| GAT | *79.3% (+13.2)* | 21.6% (+0.6) | 47.5% (+3.6) | 46.1% (+6.3) |
| GAT2 | **79.4% (+13.3)** | 21.2% (+0.2) | *49.8% (+5.9)* | *47.8% (+8.0)* |
| GraphSAGE (Random) | 67.5 (+1.4)% | *23.9% (+2.9)* | 45.1% (+1.2) | 41.9% (+2.1) |
| GraphSAGE (Neighbour) | 72.6% (+6.5) | **43.4% (+22.8)** | **62.4% (+18.5)** | **59.9% (+20.1)** |

**Vision Tasks** Table 4 demonstrates the same pattern, that the embedding function (in this case a pre-trained vision model) influences which GNN performs the best. It is interesting to note that we see that none of the GAT models achieve the highest accuracy on any of the Flickr_v2 embeddings. Instead, similar to the RoBERTa embeddings, we see that GraphSAGE performs the best.

It is important to note for VGG16 that the surprisingly poor performance of GNNs is more likely due to the large vector size of more than 25K. With a smaller embedding space better results on par with ResNet may be achieved. Of course, there is also the possibility that the embeddings provided by VGG16 are inferior to ResNet.

Table 4: Test accuracy on Flickr_v2 with different embeddings, compared against the corresponding unconnected vision model. Included is the difference $\Delta$ of each model to the unconnected model and the standard deviation of each result. The details of the embedding styles are available in Appendix B.

| Model | Embedding Styles | | |
|---|---|---|---|
| | ResNet18 | ResNet50 | VGG16 |
| Unconnected Model | 45.2% (+0.0) | *46.9% (+0.0)* | **47.0 (+0.0)** |
| GCN | 41.8% (-3.4) | 38.3% (-8.6) | *35.5% (-11.5)* |
| GAT | 38.1% (-7.1) | 37.1% (-9.8) | 27.3% (-19.7) |
| GAT2 | 42.1% (-3.1) | 41.0% (-5.9) | 34.2% (-12.8) |
| GraphSAGE (Random) | *45.4% (+0.2)* | **47.0% (+0.1)** | 35.2% (-11.8) |
| GraphSAGE (Neighbour) | **45.8% (0.6)** | 44.5% (-2.4) | 34.5% (-12.5) |

Tables 2 and 3, compared to Table 4, have far larger increases in accuracy from the best performing GNN compared to the unconnected models. This is mainly due to the fact that we were unable to fully fine-tune RoBERTa to our datasets given limited hardware and time. We hypothesise that the improvements seen in Table 2 would be smaller when compared to a fine-tuned RoBERTa. Similarly, we did not attempt to create a GraNet model using VGG16 as the results on Flickr_v2 are worse than the ResNet models and therefore for Flickr_v2 a GraNet ResNet is ideal.

Tables 2 and 3, specifically with BoW, are the only instances where GAT and GAT2 are the best models. Furthermore, the entries for BoW follow results from past papers, in contrast to the results shown on all other embeddings. With the prevalence of BoW as shown in Table 1 it begs the question as to whether we are optimising for BoW extraction rather than graph information extraction.

From these results in both language and vision tasks, we make the following key observations:

- The function $f_e$ used to extract *embeddings influences the performance of different GNNs*, so the embeddings should influence the choice of GNNs regardless of the underlying data.
- GNN models do not always outperform simple unconnected models, *graph structure is not enough* to compete against good classifiers.
- The choice of an *embedding function contributes more to the final performance* compared to the choice of a GNN model.

## 6.2 GraNet

Table 5 demonstrates that extending our standard models with graph connections provides a significant improvement. We see that these GraNet models beat the best performing GNN. The table includes two forms of GraNet models, the fine-tuned models and the trained models.

**Fine-tuned** In Flickr_v2, both GraNet and Unconnected Model use either ResNet18 or ResNet50 weights pre-trained on ImageNet (Deng et al., 2009) and then fine-tuned on the target dataset. We observe a significant increase ($+1.5\%, +1.8\%$) from the GraNet style of training compared to both unconnected models, and improvement ($1.1\%, 1.7\%$) on the best performing GNN model.

Intuitively, GraNet layers reframe graph representation learning from training a GNN on a fixed pre-extracted embedding to training both the GNN and the embedding function ($f_e$) together on the underlying data. There are obviously trade-offs between time and versatility as rather than just training a GNN the embedding function must be trained as well. But the current approach of GNN

training on images requires human, or other annotators, which is a hidden time cost. By combining the embedding function and GNN we provide a more general model that does not need external annotators and thus can work on any raw images.

**Trained from Scratch** In the Amazon datasets we do not have any pre-trained and train the GraNet model from initialised weights rather than fine-tuning. However, we still observe an improvement on both the unconnected model and best performing GNN. The improvement is smaller than in Flickr_v2 ($+0.1\%, +0.3\%$ compared to $1.1\%, 1.7\%$). But the training time is quicker ($\sim$ 1hr compared to $\sim$ 10hrs for 300 epochs).

In this case we do not have a general model as we rely on BoW embeddings for the GraNet model as well. But we can see that improvements are possible on existing GNN techniques on the embeddings themselves, without having to use large pre-trained models.

Table 5: Comparison of GraNet models against the best performing GNNs for a specific embedding. Unconnected Model refers to ResNet18 or ResNet50, in the case of Flickr_v2, and an MLP otherwise. The setup of each GraNet model is detailed in Sections 4.1 and 4.2 and the specifics of the 3 datasets can be found in Appendix B.

| Model | Flickr_v2 | | Electronics | Instruments |
|---|---|---|---|---|
| | ResNet18 | ResNet50 | Bag of Words | Bag of Words |
| Unconnected Model | 45.2% (+0.0) | 46.9% (+0.0) | 71.6% (+0.0) | 66.4% (+0.0) |
| GAT2 | 42.1% (-3.1) | 41.0% (-5.9) | *81.8% (+10.2)* | *79.4% (+13.0)* |
| GraphSAGE (Random) | 45.4% (+0.2) | *47.0% (+0.1)* | 71.3% (-0.3) | 67.5% (+1.1) |
| GraphSAGE (Neighbour) | *45.8% (+0.4)* | 44.5% (-2.4) | 76.4% (+4.8) | 72.6% (+6.2) |
| GraNet | **46.7% (+1.5)** | **48.7% (+1.8)** | **81.9% (+10.3)** | **79.7% (+13.3)** |

We also make the following key observations:

- Graph-connecting a pre-trained network *improves performance by fine-tuning feature extraction based on the graph structure*.
- GraNet models *outperform their counterparts by facilitating fine-tuning* on the graph dataset.

The architecture for each GraNet is a mixture of the best performing GNN for that embedding and the embedding extraction model ($f_e$) (a Multi-Layer Perceptron in the case of Bag of Words).

# 7 CONCLUSION

In this paper, we reveal that GNN model designs are overfitting to certain embeddings styles (*e.g.* BoW and word vectors). To demonstrate this we introduced three new datasets each with a range of embedding styles to be used as a more comprehensive benchmark of GNN performance.

We demonstrated that embedding style influences the performance of GNNs regardless of the underlying dataset. Equally, the quality of the embedding, measured by how well an unconnected baseline model performs, is a greater indicator of GNN accuracy than the GNN architecture chosen. *We therefore stress the importance of creating high quality embeddings* and *choosing the best GNN architecture based on the style of embedding* created rather than using (or trying to improve) the same GNN model for every task.

We then introduced a new approach named GraNet. *This approach allows for any large pre-trained model to be fine-tuned to a graph-connected dataset* by altering the standard message passing function. In this way we exploit graph structure information to enhance the pre-trained model performance on graph-connected datasets. We have demonstrated that *GraNet outperforms both unconnected pre-trained models and GNNs* on a range of datasets.

There is an increasing trend towards large pre-trained models and graph-connected datasets. Our work demonstrates potential pitfalls in the way GNN architectures are currently evaluated and proposes a new technique to fully exploit the benefits of pre-trained models within a GNN.

## 8 ETHICS STATEMENT

With all computer based research there is the issue of carbon footprint. Each train, validation and test run of a model requires electricity which is currently largely sourced from fossil fuels. This training cost carbon footprint is present in this picture, and attempts to keep this low by running short test on new iterations rather than long runs which could waste GPU time and energy.

## 9 REPRODUCIBILITY

We discuss how we setup all of our experiments in Section 5 including how many runs we completed for each data-point. the specific random seeds that we used for the three runs were 42, 9001 and 27032002. The number of runs for each run is also listed here and for the models that have a different number of epochs the specifics are mentioned in Appendix E.

All test runs were carried out on a single Nvidia A100 80GB GPU, we used pytorch 1.12 with cuda 11.3. All other packages where installed based on these requirements.

Appendix E also contains the specific architectures, samplers, learning rates and learning rate schedulers used in each of the model test runs. These are also linked back to Zeng et al. (2019) to provide comparable results to the ones achieved in that paper for similar datasets Flickr and Amazon.

Appendix B details the specifics of how we formed our datasets including how we downloaded the raw data, steps we took to wash the data, how we formed the graph and how we created our embeddings. The specifics of how we labelled the data and the labels that we chose are also available.

We have anonymised the dataset including the specific code used to create GraNet models, the config files used and the code to download, wash and build the datasets available here.

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
