# OpenReview forum: "Revisiting Embeddings for Graph Neural Networks"
_ICLR.cc/2023/Conference — Submitted to ICLR 2023_

### Official Review · Reviewer_r5aW · 2022-10-20

**Confidence:** 3
**Correctness:** 3
**Technical Novelty And Significance:** 1
**Empirical Novelty And Significance:** 2
**Recommendation:** 3

**Clarity, Quality, Novelty And Reproducibility:**

Clarity: The clarity is fair, please find my detailed comments in weakness 2.

Quality: The quality of this work is fair, the model is not clearly motivated (mostly the authors are just trying to make it clear what is this model doing, but do not explain why this design can be a good choice), and the results are not interpreted well (I think it would be better if the authors can explain why we get this kind of results instead of just pointing out what can we observe from experimental results).

Novelty: The novelty looks marginal to me, details are in weakness 1.

Reproducibility: Experimental setup is provided in the main paper and supplementary, and the code is also provided, therefore, I think the reproducibility should be good.

**Strength And Weaknesses:**

strengths:
1. The problem of how to combine GNNs with the unconnected models to take advantage of both is worth thinking discussing, this topic is meaningful.
2. The code is provided in the supplementary, which enhances the reproducibility of this work.

weakness:
1. Novelty of this work is not enough.
Though the topic of getting the benefit from both GNN and non-GNN models is worth exploring, the idea of combining both is straightforward. And if I understand correctly, the design of GraNet is just adding a projection step to the network input (i.e. applying some function to the input) before conducting the neighborhood aggregation and transformation steps in GNN models, and the projection step is an existing NN model (MLP/CNN), which does not have enough novelty.


2. Writing can be improved.

1) The organization of this work is not easy to follow for me.
- In the related work section, why it is important to include the paragraph "Effect of training on GNN performance"? This paragraph seems not very related to this paper.

- In the background section, why do we have Table 1? These datasets are not used in the experiments, and if the authors only want to use this to explain most of the existing datasets rely on the BoW embedding, then I don't think it needs this big table to do so.

- The background section only has one subsection (3.1), then why do we need this separate subsection? And the paragraphs are more than just notations, so I don't feel it is proper to put things under the subsection "Notations".

- The method section is slightly confusing for me. First, the model itself is not clearly introduced, it needs some effort to figure out how many components are there in this model, and how is each component designed and how is the model trained. Second, it looks to me that the two subsections are the instantiation of the proposed model, I don't feel it needs these 2 long paragraphs for instantiation.

- For the evaluation section, first, the titles are not informative, especially the "GraNET" one (which is actually not talking about GraNET, but is the main result of comparing GraNET against different baseline models). Second, it would be better if the author can provide richer results (like ablation studies) instead of using long paragraphs to give a long but not concise statement for the observations which we can clearly find in the table.

2) The motivation for this work is not clearly delivered.
- mostly the authors are just trying to make it clear what is this model doing, but do not explain why this design can be a good choice

3) The experimental results are not explained with enough thought and analysis.
- I think it would be important to explain why we get this kind of results instead of just pointing out what can we observe from experimental results

4) There are a few typos:
- page 4 first line, "architecturwa" -> "architectures"
- page 5 last line, "Equation (6)" -> "Equation (5)".




**Summary Of The Paper:**

This work aims to examine the quality of different types of input embeddings for GNNs, and design a model architecture by combining both GNNs and non-GNN models (which are referred to as unconnected models in the paper). The authors provide empirical results to demonstrate the importance of having a high-quality input embedding, then propose the GraNet model to combine GNNs and non-GNN models, and finally provide 3 newly-processed datasets to justify the proposed model.

**Summary Of The Review:**

My major concern with this work is, the contribution of this work is limited and the writing is hard to follow for me. The authors provide some empirical results to demonstrate that it is important for GNN layers to get a good input feature but does not really make it clear why and when will each type of input feature leads to good or bad performance. The proposed GraNet is not well-motivated and if I understand correctly, the model is just adding a feature mapping function on the input of the GNN, which is not a surprising idea. Therefore, I think this paper is not ready to get published in this venue at this time and needs further improvement.

---

### Official Review · Reviewer_rePb · 2022-10-23

**Confidence:** 3
**Correctness:** 3
**Technical Novelty And Significance:** 2
**Empirical Novelty And Significance:** 2
**Recommendation:** 5

**Clarity, Quality, Novelty And Reproducibility:**

The paper is not particularly clear in terms of contribution and novelty. There are no problems of reproducibility thanks to the information provided in the paper, supplementary materials and source codes.

**Strength And Weaknesses:**

- Weaknesses:
-- it's not clear to me what is the main goal or contribution of the paper. For instance, the abstract starts talking about the exploration and analysis of different embedding techniques in GNNs. Later, it's claimed that a new type of layers is proposed, and even, as first contribution (included at the end of the Introduction) the proposal of "new datasets and a rich set of accompanying embeddings to better test the performance of GNNs" is mentioned.
-- writing should be reviewed to make the reading less hard for the reader.

- Strengths: quantitative results are very promising.

**Summary Of The Paper:**

The paper explores different embedding extraction techniques in Graph Neural Networks (GNNs), and proposes Graph-connected Network (GraNet) layers, claiming that this approach improves the accuracy compared to traditional GNNs.

**Summary Of The Review:**

The main goal and contribution of the paper is confusing: is this a comprehensive analysis of embeddings for GNNs, the proposal of a new GNN model/approach, or the proposal of "new datasets and a rich set of accompanying embeddings to better test the performance of GNNs"?

There are statements that are not clear: for instance, in the abstract, the authors state that "As an alternative, we propose Graph-connected Network (GraNet) layers to better leverage existing unconnected models within a GNN. Existing language and vision models are thus improved by allowing neighbourhood aggregation. This gives a chance for the model to use pre-trained weights". This statement is far from trivial in my opinion and it would need further justification. Why the proposal made by the authors increases the chances of using pre-trained weights?

I hesitate about the technical and methodological novelty of the GraNet approach. For instance, what is the main originality and contribution with respect to Brody et al. (2021)? What are the main intuitions behind GraNet? A better visual description of GraNet and its comparison with prior art will be very welcome. In fact, to be totally honest, I'm not sure I fully understand the merit, motivation and rationale behind their proposal.

---

### Official Review · Reviewer_mwDM · 2022-10-24

**Confidence:** 4
**Correctness:** 3
**Technical Novelty And Significance:** 2
**Empirical Novelty And Significance:** 2
**Recommendation:** 3

**Clarity, Quality, Novelty And Reproducibility:**

Clarity. Overall, this paper is easy to follow. However, there are some presentation issues.
- some parts are not clear. For example, where the prior knowledge (about which GNNs perform better on cerntain data) is from has not been introduced. Some experimental settings are not clear such as train-test split.
- in section 3.1, some concepts, e.g., fine-tuning and freezing, can be moved to the appendix or even removed.

Quality. The quality of this work is below average.

Novelty. Technically, the novelty of this paper is limited because the proposed framework is a straightforward combination of neural networks and GNNs.

Reproducibility: The source code is provided for reproducibility.

**Strength And Weaknesses:**

Strengths:
- The problem of investigating the influence of embedding for GNN is very interesting and there are rare studies on this problem.
- Experiments on different types of datasets with different settings (embeddings) have been conducted to validate the effectiveness of the proposed framework.

Weakness:
- The novelty of this paper is limited. The proposed framework is a straightforward combination of neural networks and GNNs.
- Presentation in this paper, especially introducing the method, sometimes is not very clear: in Sections 4.1 ad 4.2, the authors mentioned that "GraphSAGE performed the best on ResNet embeddings" and "we also find that GAT models perform the best on this task", I wonder where are these conclusions from.
- It seems that the proposed method is not generalized enough. For different types of datasets introduced in this paper, selecting which GNN architecture is based on prior knowledge (the claims above). A question is how should one select the appropriate GNN architecture if a new dataset is used.
- For the dataset construction, since the unconnected model can already achieve comparable or even better performance than GNN such as GAT in Flickr_v2 data, does it mean that the structure in this data is unnecessary?
- In the experiments, there are some issues:
1) Details of the data split are not provided.
2) Although for different datasets, different GNNs are used to demonstrate the effectiveness of the proposed framework. It will be interesting to show the results of mixing other GNNs with certain unconnected models. Such comparison can better verify the effectiveness of the proposed combination strategy.

**Summary Of The Paper:**

This paper studies how the quality of embedding affects the performance of GNNs. The authors selected two types of data, images and texts, and tested different embedding extraction techniques. Then a general framework Graph-connected Network (GraNet) is proposed to combine GNNs and unconnected models to better learn embeddings of input data. Experiments on both images and texts with different embedding methods have been conducted to demonstrate the effectiveness of the proposed framework.

**Summary Of The Review:**

This paper studies an interesting and important problem. Overall this paper is well-organized and easy to read. Experiments on different types of datasets have been conducted. However, I have some concerns about the novelty, and experiments, as well as the presentation of this paper. Therefore, I would like to reject this paper.

---

### Official Review · Reviewer_FPbc · 2022-10-25

**Confidence:** 3
**Correctness:** 2
**Technical Novelty And Significance:** 1
**Empirical Novelty And Significance:** 2
**Recommendation:** 3

**Clarity, Quality, Novelty And Reproducibility:**

Reading the abstract and introduction, it appears that the paper addresses the general issue of graph embeddings and their impacts on the performance of GNN architectures. However, the paper is really about a very specific setting: namely,  it assumes that the "nodes" in the so-called "graph"  datasets under considered are either images and texts; and they are connected in  some form of "graph structures", e.g., based on “Co-viewed”, “Co-bought” and “Similar Items” (for the Amazon datasets).  The "(node) embeddings" the paper talks about are essentially some CNN networks or MLP networks that are "pre-trained" on the images or texts associated with the nodes, and which are then used as "node features".    The "graph structures" on the nodes considered (based on the descriptions of the datasets) are rather "secondary" in that they are comments, labels provided by humans or activities associated with humans, thus rather "weak" or "ad hoc" in a sense. It is no wonder that the authors find most GNNs do not necessarily provide additional improvements over the "fine-tuned" models without utilizing the graph structures.


I find the description of the  proposed GraNet framework is rather confusing and needs better explanation and justification. For example,  I don't understand how the light blue regions work, in particular, with respect to how the forward pass of the current $h_i$  and the forward pass of its neighbors mean. You also mentioned about "The light orange region and resulting channel stacks, B, represent the graph-based Message Passing stage where the new representations are altered, aggregated and finally combined
with the current node representation, following the description in Equation (3).  How are they altered, aggregated and combined? Or are you simply stacking a GNN on the "embeddings"? Are these done in a layer by layer fashion, or only at the last stage (i.e., taking the outputs of a CNN applied to the images associated with the nodes)?

Why do you think your findings can be generalized to other datasets (where nodes are not images or texts)?



**Details Of Ethics Concerns:**

No ethical issues.

**Strength And Weaknesses:**

Strength:

 + Exploring  the impact of (pre-trained) embeddings on the performance of different GNN architectures.

 Weaknesses:

  - The paper is rather poorly written and claims more than what is actually accomplished.

  - The proposed GraNet framework is difficult to understand and needs better explanation and  justification.



**Summary Of The Paper:**

The paper considers the impact of (pre-trained) embeddings on the performance of different GNN architectures. The datasets under considered are either images and texts with some form of "graph structures", e.g., based on “Co-viewed”, “Co-bought” and “Similar Items” (for the Amazon datasets). The authors fund that only some GNN models yield  an improvement in accuracy compared to the accuracy of
models trained from scratch or fine-tuned on the underlying data without utilizing the graph connections. As an alternative,the authors propose Graph-connected Network (GraNet) layers to better leverage existing unconnected models within a GNN, and show 7.7% and  1.7% improvements on Flickr v2, GraNet over  GAT2 and GraphSAGE.

**Summary Of The Review:**

The paper aims to investigate the impact of (pre-trained) embeddings on the performance of different GNN architectures. However, both the problem considered and the method proposed are really specific to datasets where nodes are images or texts that are "connected" via some (weak) forms of connections. The findings are hard to interpret, and they do not shed light on the performance of general GNNs.

---

### Decision · Program_Chairs · 2023-01-20

**Decision:**

Reject

**Justification For Why Not Higher Score:**

No rebuttal provided.

**Justification For Why Not Lower Score:**

N/A

**Metareview: Summary, Strengths And Weaknesses:**

This paper is concerned with investigating the quality of graph neural network (GNN) embeddings and their impact on the overall GNN performance. The study is conducted by considering text and image data. Further, the authors investigate the impact of pre-trained representations on GNNs.

The reviewers believe that the problem considered here is an important one and work in this area should be useful.

However, all four expert reviewers agree that this paper is not ready for publication in ICLR. There is a key concern around the presentation of the paper, including the overall writing and explanation of results and how these support the claims. The reviewers also believe that the contribution is limited: a projection step is added to the GNN but this is a simple extension (likely already used by practitioners) that seems to make some difference in the considered cases. Could there additionally be some theoretical or more extensive analysis to understand whether this trick is more widely applicable? Essentially, the reviewers are questioning whether the material in this work is enough to be used as a basis for further research. I also note that there has been no rebuttal after the review period.